

# Discrepant long-term nitrogen mineralization in soil at early and later period after fertilization

Hongqin Zou [1], Minggang Xu [1,2], Keyu Ren [1], Dejin Li [1], Wenju Zhang [1], Changai Lu [1], Yinghua Duan [1*]

[1] State Key Laboratory of Efficient Utilization of Arid and Semi-arid Arable Land in Northern China / Institute of Agricultural
Resources and Regional Planning, Chinese Academy of Agricultural Sciences, Beijing, 100081, China.

[2] Engineer and Technology Academy of Ecology and Environment, Shanxi Agricultural University, Taiyuan, 030031, China.

[*] *Correspondence* to: Yinghua Duan (duanyinghua@caas.cn)

**Abstract.** Soil mineralization, the process of organic to inorganic N which balances the N uptake by crop and N loss to
environment, was always quantified by short-term (one day to thirty weeks) incubation experiment. However, the residuary
effect of fertilization for N mineralization, especially manure application, is long-term existed, which is important and thought
to be shaped by fertilization, soil properties, and climate. Here, we defined and examined dynamic shifts in long-term N
mineralization (LT-$N_{min}$) between the first decade and later period after treated with no-fertilizer, conventional fertilizer
with/without manure, fertilizer with stover return, in six typical agricultural zones over multiple decades. Soil total N (TN)
and available N (AN) increased at rates of 10.1–58.2 mg· kg$^{-1}$·yr$^{-1}$ and 1.41–4.13 mg· kg$^{-1}$·yr$^{-1}$, respectively, by manure
application at five sites, suggesting that manure enhanced both soil N storage and availability. The LT-$N_{min}$ rate, defined as
slope of the correlation between soil AN and TN indicated that AN increased by 70–81 mg per gram of TN increase, regardless
of fertilization. Considering the fertilization period, the LT-$N_{min}$ rate with manure application were higher in the first decade
(42–181 mg·g$^{-1}$) than those in later period (33–92 mg·g$^{-1}$) at all sites. Variance partitioning analysis showed that soil properties
contributed 35% to LT-$N_{min}$ in the first decade and increased to 45% in the later period, while climatic conditions contributed
19% first, but 8% in the later period. Structural equation modeling suggested that LT-$N_{min}$ was directly affected by annual
temperature, with a standardized path coefficient of 0.86 in the first decade and 0.45 in the later period. These results showed
that the residual N in soil was mineralized with a high rate in the first decade after fertilization and then slowed down, and the
interactions between climate and soil had an enhanced impact on LT- $N_{min}$ in later years of fertilization. This context-dependent
understanding of interactions between soil properties, N cycling, and climate can thus inform soil management strategies to
improve N availability and reduce the N loss to environment.

**Keywords.** Nitrogen mineralization · Climatic effect · Soil properties · Long-term fertilization · Manure application

## 1    Introduction

Nitrogen (N) is an essential soil nutrient required for plant growth and crop production. Although more than 90% of N in soil
is present as organic N (Schulten and Schnitzer, 1997), plants primarily take up inorganic N and only absorb a small proportion



of low molecular weight organic N compounds (*e.g.*, glycine), typically under extreme conditions (Nasholm et al., 1998).
Enhanced soil N mineralization provided the opportunity to increase the N fertilizer use efficiency, enabling sufficient crop
yields with reduced risk in N loss (Giacomo et al., 2012). Therefore, soil N mineralization ($N_{min}$) from organic forms to labile
or biologically available ammonia or nitrates can greatly impact crop production and agroecosystem sustainability.

Soil $N_{min}$ rates were usually determined by short-term intermittent leaching or isotopically labeled laboratory incubation
experiments (Stanford and Smith, 1972; Zhang et al., 2019), which represented the soil capacity of N mineralization at a certain
state in the short term. Actually, soil N-supplying capacity increased with the continuous fertilization, and fertilizer-N retained
in the soil may contribute substantially to crop N uptake in subsequent years (Proffenbarger et al., 2018; Vonk et al., 2022),
indicated that the effects of fertilization are long-lasting. Therefore, estimating site-specific $N_{min}$ rates by incubation experiment
could pose several challenges (Wade et al., 2018). For instance, estimation of soil $N_{min}$ performed in the laboratory
(Beesigamukama et al., 2021) might fail to account for the differences in the effects of long-term fertilization that occur under
field conditions (Risch et al., 2019). Moreover, climatic conditions and fertilization regimes shaped soil properties over long
time scales (Doetterl et al., 2015; Wang et al., 2018), and thus understanding soil long-term $N_{min}$ (LT-$N_{min}$) under various
climatic conditions, soil types, and fertilization practices is crucial for reliable and robust estimates of soil N availability, and
ultimately, productivity in terrestrial ecosystems.

Climatic conditions and soil properties have been identified as primary factors controlling $N_{min}$ on a global scale (Liu et al.,
2017; Risch et al., 2020). The rate of $N_{min}$ increased with mean annual temperature (Dawes et al., 2017) and precipitation
(Burke et al., 1997). Soil total N (TN) content could be an informative predictor of $N_{min}$ since the inorganic mineralization
substrates were included in this pool (Dessureault-Rompre et al., 2015; Matar et al., 2008). However, other studies have shown
that $N_{min}$ may share only a weak correlation with TN (Vigil et al., 2002). Soil properties could mutually influence each other
(Kleber et al., 2015) and thus an integrative understanding of the impacts of various climate and soil properties on $N_{min}$ is
essential for predicting crop performance under different climatic conditions and management practices, especially in the
context of global warming.

In addition to climate and soil properties, fertilizer resources could directly affect soil N mineralization rates. For example,
various fertilizer applications were well-known to profoundly impact soil microbial biomass and bacterial ureolytic and
chitinolytic communities, which in turn drive N mineralization (Lee and Jose, 2003; Treseder, 2010; Yang et al., 2019). Manure
application and stover return have also been shown to improve both $N_{min}$ potential and rates by providing carbon and nutrient
substrates that enhance enzyme activity, such as that of urease (Qin et al., 2013). Some studies have found that manure
application increase the proportion of easily degraded soil organic matter and thus the content of potentially mineralized soil
N (Nett et al., 2009). Fertilization not only affected soil $N_{min}$ directly by providing metabolic resources, but also altered $N_{min}$
by influencing other soil properties. For instance, manure application promoted soil N mineralization by increasing the contents
of soil organic carbon (SOC), soil micro-biomass nitrogen (SMBN), and by enriching for bacteria (*Gemmatimonadetes* and
Latescibacteria), while conventional urea application had no effect on $N_{min}$ (Wei et al., 2011; Guo et al., 2019). However, the
different roles of fertilization, climate and soil-related factors in N mineralization are not well understood.



This study was designed to identify the long-term effects of fertilization and climate conditions on soil N availability, and to guide development of the best N management practices under diverse agricultural conditions. Using six 21–29 years long-term agricultural experiment sites in China, dynamic shifts in soil TN and available N (AN) were quantified to determine the long-term N mineralization (LT-$N_{min}$) rate, defined as the ratio of AN change rate to TN change rate. The LT-$N_{min}$ rate thus largely reflects soil $N_{min}$ during the study period, since it can show how much of the TN accumulated in soil is activated to AN annually. The specific objectives of this work were to: (1) investigate the soil LT-$N_{min}$ under fertilization with manure or stover return; (2) examine the soil LT-$N_{min}$ over different durations of fertilizer application; and, (3) evaluate the effects of fertilizer treatment, climate, and soil properties on soil LT-$N_{min}$.

## 2    Materials and methods

### 2.1    Site descriptions

Six long-term experiments were studied at Gongzhuling (GZL), Zhengzhou (ZZ), Urumqi (UM), Yangling (YL), Zhangye (ZY), and Qiyang (QY). The GZL is located in the Northeast; ZZ is in central China; UM, YL, and ZY are in the Northwest; and QY is in the south of China (Fig. 1). All the experiments were set up in 1990 except ZY, which was initiated in 1982. The climates in the locations ranged from arid, temperate zones to humid subtropical zones (Fig. 1). Annual mean temperature varied from 4.5 °C at GZL to 18.0 °C at QY. Annual precipitation ranged from 127 mm to 1 250 mm, and annual evaporation ranged from 975 mm to 2570 mm. Frost-free period varied from 135 days at GZL to 300 days at QY. The annual active accumulated temperature (daily temperature > 10°C), ranged from 2 800 °C at GZL to 5 600 °C at QY (data from China meteorological sharing service system, http://cdc.cma.gov.cn/).

All experimental fields were used for agriculture for more than ten years before the experiments were initiated. Soil classifications and properties before the treatments (1982 or 1990) were presented in Table S1. The initial SOC content was higher at GZL and ZY sites (11.5–13.2 g·kg$^{-1}$) than other sites (6.7–8.8 g·kg$^{-1}$). The highest total N content was detected at GZL (1.40 g·kg$^{-1}$), follow by QY (1.07 g·kg$^{-1}$), and other sites (0.67–0.87 g·kg$^{-1}$). The soil C/N ratio was the highest at ZY (13.4), and the lowest at QY (7.4). The soil pH was 5.7 at QY and 7.6-8.6 at other sites.



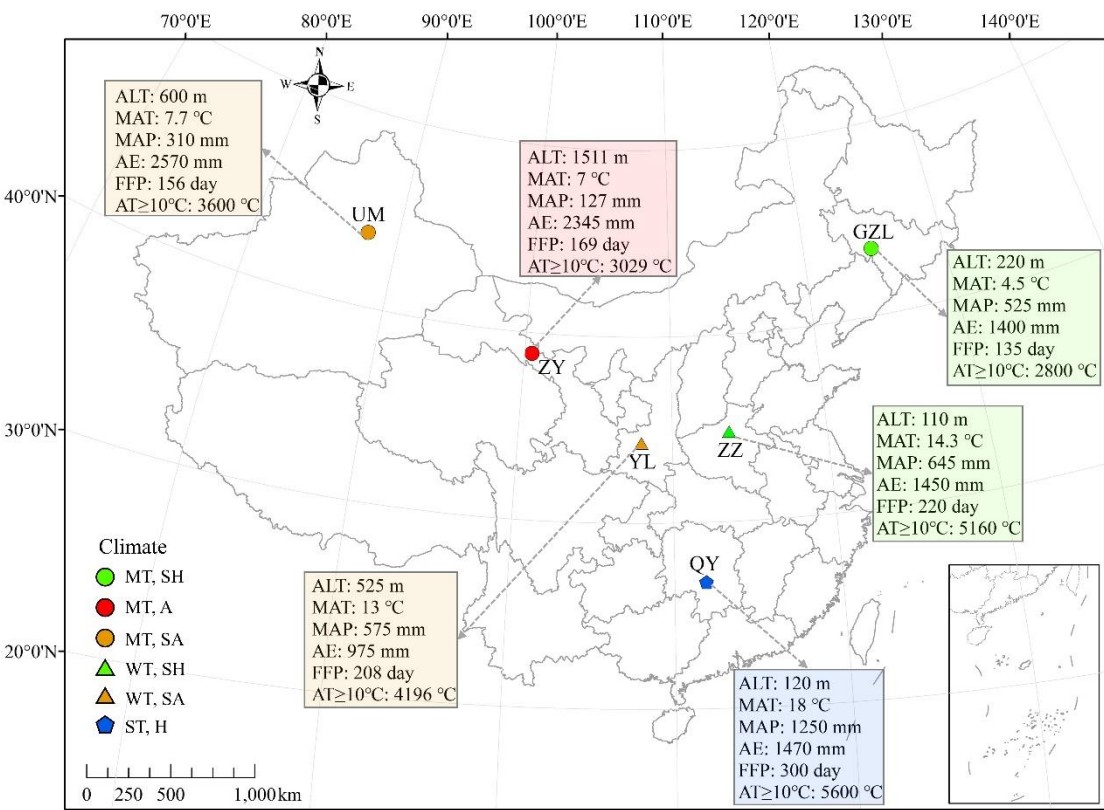

**Figure 1: Locations and climatic conditions of the six long-term experimental sites.** GZL, Gongzhuling; ZZ, Zhengzhou; UM, Urumqi; YL, Yangling; ZY, Zhangye; QY, Qiyang. MT, mild-temperate; WT, warm-temperate; ST, subtropical-temperature; A, arid; SA, semi-arid; SH, semi-humid; H, humid; ALT, altitude; MAT, mean annual temperature; MAP, mean annual precipitation; AE, annual evaporation; FFP, frost free period; AT ≥ 10℃, accumulated temperature ≥ 10℃.

## 2.2 Cropping systems

Wheat (*Triticum Aestivium* L.) or maize (*Zea mays* L.*)* were grown alone or in rotation at all sites. The GZL, UM, and ZY sites were used for mono-cropping, while rotating crops were grown at ZZ, YL, and QY sites (Table S1). Maize was sown during late April to early May, and harvested during late September to early October under mono-cropping practices. Wheat (spring wheat) was seeded in late March and harvested in late July at ZY. At UM, spring wheat was sown during late March to early April and harvested in mid-July, winter wheat was sown in late September in the same year and harvested during late June to early July in the next year. In double cropping sites, maize was sown in mid-June and harvested between late September and early October at ZZ and YL, or sown in late March to early April and harvested in mid-June at QY. Wheat (winter wheat) was sown in mid-October and harvested during mid- to early June at the ZZ and YL sites, whereas it was sown in early November and harvested in mid-May at QY. Each field was under the same rotation for 2–3 years prior to initiating the experiment at each site.



## 2.3 Field design and treatment

There were five treatments at each study site: unfertilized control (CK); inorganic N, phosphorus and potassium (NPK);
inorganic NPK plus manure (NPKM); 1.5- or 2-times application rate of NPKM (hNPKM); and inorganic NPK with stover
return (NPKS); except ZY where only CK, NPK, and NPKM treatments were available (Table S2).

At GZL, ZZ, YL and QY, the total amount of N applied was the same for all fertilizer treatments (except CK and hNPKM),
*i.e.*, 30% of the total N applied was inorganic while the rest was derived from composted manure. Both the inorganic N fertilizer
and manure application rates were 1.5 times higher in hNPKM treatments than in NPKM at the above four sites. At UM, the
manure supplied in NPKM was 123% more N than that present in the NPK component, while the inorganic N and manure
application rates for hNPKM were two times that in the NPKM treatment. At ZY, the amount of inorganic N applied was same
for all fertilizer treatments (except CK), and the N provided by the manure component of NPKM was equivalent to the N rate
present in NPK. For NPKS treatments, crop stover was chopped and incorporated into the soil *in situ* immediately following
harvest, annually. All the stover harvested from each NPKS plot was returned to the same plot. Nutrients (N, P, and K) in the
stover, however, were not counted towards N applications because changes in their availability with time were not recorded.

The inorganic N, P and K fertilizers applied were urea, calcium superphosphate, and potassium sulfate, respectively. The source
of manure was from household livestock such as pig, cattle, horse, and goat. Manure was applied before seeding once a year
for all sites with mono-cropping systems. At sites with double cropping, manure was applied before wheat seeding. The
fertilizer treatments were arranged in a completely randomized design in the field. There were three replicates at ZY and ZZ
with a plot size of 33 m$^2$ and 43 m$^2$, respectively. Due to the relatively large size of each plot (196–468 m$^2$), replications were
not included when the experiment was established for demonstration/outreach purposes at the GZL, UM, YL and QY sites.
These plots were isolated by 100 cm cement baffle plates.

To ensure high maize yields, the target fertilizer application rates were slightly different at the six sites based on the common
practices of local growers, which were specific to local cultivars, climate, and soil conditions (Table S2). The annual target
nutrient application rates ranged from 120 kg· N ha$^{-1}$–300 kg· N ha$^{-1}$ for the mono-cropping systems (NPK at GZL, UM, and
ZY). The N application rates were 188–210 kg· N ha$^{-1}$ for maize and 90–165 kg· N ha$^{-1}$ for wheat in the double cropping
rotation systems (ZZ, YL, and QY). The P and K fertilizer application rates were 36–150 kg· P$_2$O$_5$ ha$^{-1}$ and 61–150 kg· K$_2$O
ha$^{-1}$ for the mono-cropping systems, respectively, every crop season. The P and K fertilizer application rates were 36–198
kg· P$_2$O$_5$ ha$^{-1}$ and 36–124 kg· K$_2$O ha$^{-1}$ for the double cropping systems, respectively. One-third of the N fertilizer was applied
as basal fertilizer before seeding and the rest applied as topdressing at the jointing stage at GZL. For ZZ and UM sites, 60%
of the N fertilizer was applied as basal fertilizer before seeding and 40% as topdressing at the jointing stage. At ZY, YL and
QY, half of the N fertilizer was applied as basal fertilizer before seedling and the remaining was applied as a topdressing during
the growing season. All the P and K fertilizers were applied as basal fertilizers before seeding at all sites.





### 2.4 Soil sampling and analysis

The data in this study was collected from 1982–2002 at ZY, and from 1990 to 2011, or 2012, or 2014, or 2018 in other sites (Table S2). While soil samples were not available for all sites or for every year; the quantity of available data are shown in Table 1. Soil samples were collected from the topsoil (0–20 cm) after harvest in early October. At ZY and ZZ, five core (2.5 cm i.d.) samples were randomly collected and mixed as one composite sample for each replicate plot. For the other sites, a total of 20 soil core samples (0–20 cm depth), were collected from each treatment plot using a 2.5 cm diameter auger and five

core samples were mixed as one composite sample, totaling four samples for determining soil properties. The soil samples were then air-dried, sieved through a 2 mm screen to determine pH (1:2.5 w/v water) and 0.25 mm mesh for soil nutrient contents. The average values were used for statistical analysis.

Soil organic carbon (SOC) content was determined by vitriol acid-potassium dichromate wet oxidation (Walkley, 1935). Soil TN was determined by the Kjeldahl digestion-distillation method (Black, 1965). Soil total P was determined by the

145 molybdenum blue colorimetric method using an $HClO_4$–$H_2SO_4$ solution for digestion (Murphy and Riley, 1962). Soil total potassium and available potassium was determined by flame photometer (Lu, 2000). Soil AN was determined by alkaline solution diffusion (Lu 2000), and available P was quantified by the Olsen method (Olsen, 1954).

Climatic data, including mean annual precipitation (MAP), mean annual temperature (MAT), mean annual humidity and mean annual evaporation, were obtained from the China Meteorological Data Service Center (http://cdc.cma.gov.cn/).

### 2.5 Estimation of soil TN and AN rate of change and LT-$N_{min}$

To analyze the cumulative effect of the fertilizer regime on soil TN, AN and long-term N mineralization (LT-$N_{min}$), the annual rates of change for soil TN (g·kg$^{-1}$·yr$^{-1}$) and AN (mg·kg$^{-1}$·yr$^{-1}$) were determined using the least square linear regression (Tang et al., 2008).

$$TN = a_1 + b_1 X \tag{1}$$

$$AN = a_2 + b_2 X \tag{2}$$

Where $TN$ was total N content (g·kg$^{-1}$), $AN$ was available N content (mg·kg$^{-1}$), $a_1$ and $a_2$ are the intercepts (*i.e.*, the initial TN and AN), $b_1$ and $b_2$ were the slopes (annual rates of change for TN and AN) and $X$ is the year.

The soil TN and AN content were positively correlated, and the slope of the linear relationship between AN and TN was used to calculate the annual increase in AN per 1 g· TN ha$^{-1}$ increase annually. This slope was defined as the soil long-term N

mineralization rate (LT-$N_{min}$) here.

$$AN = a_3 + LT{-}N_{min} * TN \tag{3}$$

The soil C/N, C/P and N/P ratios were calculated from the ratio of SOC, soil TN, and total P contents (Qaswar et al., 2019).



## 2.6 Data analysis

Linear regression analyses were conducted using the Sigmaplot 10.0 system (Systat Software, Inc. SigmaPlot for Windows
2006). Variation partitioning analysis (VPA), to partition the variance shared by all factors, was used to quantify the unique
contribution of each group of factors including soil properties, fertilizer treatments, and climatic conditions (Legendre, 2007).
A negative value in the variance explained for group factors was interpreted as zero, which indicated that the explanatory
variables explained less of the variation than random normal variables (Delgado-Baquerizo et al., 2017). The VPA was
conducted with the R package vegan v.3.2.4 (R Development Core Team, 2016).

Structural equation modelling (SEM) was further used to evaluate the direct and indirect relationships between long-term N
mineralization rate and climatic conditions, soil properties, and/or fertilizer treatments. This approach could partition isolate
the direct and indirect effects that one variable may have on another (Grace, 2006; Hershberger, 2001) and was therefore
helpful for exploring complex relationships in natural ecosystems. Owing to strong correlations among the factors within each
group, we conducted principal component analysis (PCA) to create a multivariate functional index before construction of
structural equation models (Chen et al., 2019). The first component (PC1), which explained 40.93–98.67% of the total variance
for these three groups, was then introduced as a new variable to represent the combined group properties into the subsequent
analysis (Table S3). The fitness of the final model was evaluated using the model $\chi^2$ test ($p > 0.05$) and the root mean-squared
error (RMSEA, $< 0.05$) of approximation. The structural equation modelling analyses were conducted using AMOS 21.0
(Amos Development Corporation, Chicago, IL, USA).

## 3 Results

### 3.1 Increased soil TN and AN by long-term manure application

To better understand the effects of long-term diverse fertilization on soil nitrogen (N) storage and availability, annual rates of
change (ARC) for soil total N (TN) and available N (AN) were examined at six sites with long-term organic or inorganic
fertilizer treatments (Table 1). Linear regression analyses showed that NPK manure (NPKM) or high manure (hNPKM)
applications resulted in significantly increased soil TN contents over time at all sites except QY. The ARC of TN was higher
in hNPKM treatments at different sites (range = 28.8–58.2 mg·kg$^{-1}$·yr$^{-1}$) compared with that in NPKM treatments at the same
sites (range = 10.1–48.3 mg· kg$^{-1}$·yr$^{-1}$). Positive ARC values were also observed in sites treated with stover return (NPKS) at
the GZL, ZZ, and YL sites (range = at 8.50–18.0 mg·kg$^{-1}$·yr$^{-1}$). By contrast, significant increases in TN were only observed at
ZZ and YL sites under inorganic fertilizer (NPK) treatments, while TN remained constant at other sites. Notably, TN declined
in the unfertilized control (CK) treatment (-18.0 mg· kg$^{-1}$· yr$^{-1}$) at the UM site, but remained stable at other sites. At QY, no
significant changes in TN were observed over the 23-year treatment period. Comparison of sites with positive ARC values
under NPKM and hNPKM treatments indicated that the increase was highest at GZL, followed by UM and YL, and lower at
the ZZ and ZY sites.




The dynamics of soil AN followed similar pattern as those observed for TN (Table 1). Specifically, ARC values for soil AN
increased in both NPKM and hNPKM treatments at all sites except QY (range = 1.41–4.13 mg·kg⁻¹yr⁻¹) over the experimental
treatment period. Similarly, soil AN also increased in NPKS treatments at the ZZ and YL sites (ARC = 1.52 mg·kg⁻¹yr⁻¹ and
2.28 mg·kg⁻¹yr⁻¹, respectively), but decreased at QY (ARC = 1.32 mg·kg⁻¹yr⁻¹), and kept constant at the GZL and UM sites.
Similar to soil TN, AN also significantly increased in NPK treatment plots at ZZ and YL. In the unfertilized CK plots, AN
decreased at UM (ARC = -0.97 mg·kg⁻¹·yr⁻¹) and QY (ARC = -0.91 mg·kg⁻¹·yr⁻¹). These results indicated that long-term
application of manure led to higher levels of soil N compared to treatments with inorganic fertilizer.

**Table 1: Annual rates of change (ARC, mg·kg⁻¹·yr⁻¹) of soil total nitrogen (TN) and available nitrogen (AN) under various long-term treatments at the six study sites.**

| Sites | Experimental periods | $n_1$ | CK | NPK | NPKM | hNPKM | NPKS |
|---|---|---|---|---|---|---|---|
| | | | | Annual rates of change of soil TN | | | |
| GZL | 1990–2018 | 27 | 0.00 | 0.70 | 48.3** | 58.2** | 8.50** |
| ZZ | 1990–2011 | 21 | 0.43 | 6.73* | 17.9** | 28.8** | 13.4** |
| UM | 1990–2011 | 22 | -18.0** | -5.74 | 23.8** | 32.3** | -11.6 |
| YL | 1990–2014 | 20 | 3.19 | 15.3** | 30.4** | 33.7** | 18.0** |
| ZY | 1982–2002 | 11 | -3.16 | 4.86 | 10.1** | NA | NA |
| QY | 1990–2012 | 22 | 2.13 | -3.65 | 2.82 | 10.5 | -6.72 |
| | | $n_2$ | CK | NPK | NPKM | hNPKM | NPKS |
| | | | | Annual rates of change of soil AN | | | |
| GZL | 1990–2012 | 18 | -0.61 | -0.50 | 2.32** | 3.02** | 0.47 |
| ZZ | 1990–2011 | 21 | 0.27 | 0.81** | 1.41** | 2.23** | 1.52** |
| UM | 1990–2011 | 22 | -0.97** | 0.52 | 1.75** | 3.63** | 0.68 |
| YL | 1990–2009 | 9 | 0.56 | 1.05* | 4.13** | 3.65** | 2.28** |
| ZY | 1982–2002 | 11 | 0.18 | 0.72 | 2.01** | NA | NA |
| QY | 1990–2012 | 22 | -0.91** | -0.70 | 0.56 | 1.07 | -1.32** |

Notes: The change rate was determined using a least squares linear regression: Y = a + bX, where Y is the soil TN or AN; X is the year; b is the slope (*i.e.*, the change rate). GZL, Gongzhuling; ZZ, Zhengzhou; UM, Urumqi; YL, Yangling; ZY, Zhangye; QY, Qiyang. CK, unfertilized control; NPK, inorganic N, phosphorus and potassium; NPKM, inorganic NPK plus manure; hNPKM, high application rate of NPKM; NPKS, inorganic NPK with stover return. $n_1$ and $n_2$, quantity of the available annual data of soil TN and AN content, respectively. NA, data not available because the treatments are not included. *, significant correlation at $P < 0.05$. **, significant correlation at $P < 0.01$.

### 3.2 Responses of soil long-term N mineralization to fertilization treatments

In order to characterize differences in the factors contributing to shifts from TN to AN under organic and inorganic fertilizers,
we next examined the long-term N mineralization (LT-$N_{min}$) rate, *i.e.* the slope of the linear correlation between AN and TN,
at each site. We found a highly significant positive correlation ($p < 0.01$) between soil AN and TN under all treatments across
sites (Fig. 2). The LT-$N_{min}$ rates for the CK, NPK, NPKM, hNPKM, and NPKS treatments were 76.1, 81.1, 70.3, 74.2, and
76.9 mg·g⁻¹, respectively, which suggested that around 70–80 mg N could be transformed to available N for every 1 g·kg⁻¹
increase in TN, and the LT-$N_{min}$ was generally similar among these strategies over long-term application.

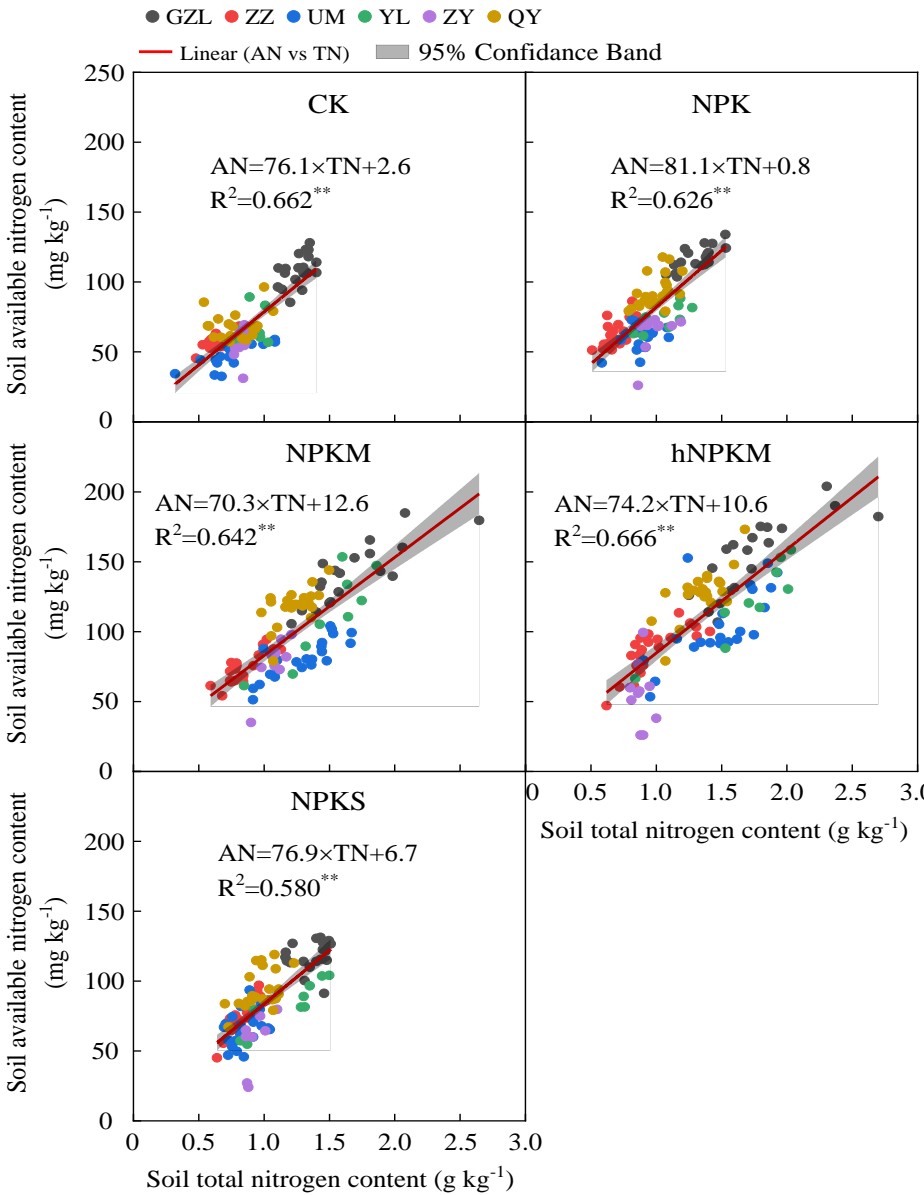

**Figure 2: The linear relationship between soil AN and TN content under various long-term treatments (n = 104).** The slope of the linear relationship between AN and TN was used to calculate the annual increase in AN per 1 gram· TN ha-1 increase annually. GZL, Gongzhuling; ZZ, Zhengzhou; UM, Urumqi; YL, Yangling; ZY, Zhangye; QY, Qiyang. **, significant correlation between the soil AN and TN at P < 0.01. CK, unfertilized control; NPK, inorganic N, phosphorus and potassium; NPKM, inorganic NPK plus manure; hNPKM, high application rate of NPKM; NPKS, inorganic NPK with stover return.

### 3.3 Differences in soil LT-$N_{min}$ between the first decade and later period of fertilizer application and the factors influencing these changes

We next sought to determine whether the effects of manure or conventional fertilizer changed over time by comparing LT-$N_{min}$ rates between the first decade and the later period of treatment at each site (Fig. 3). Linear regression analyses indicated that





both the NPKM and hNPKM treatments resulted in significantly ($P < 0.05$ or $0.01$) increased soil AN, concurrent with significantly enhanced soil TN, at all sites except YL and ZY. Notably, LT-$N_{min}$ rates in soils treated with manure were higher in the first decade (range = 42–181 mg·g$^{-1}$) than in the later period (range = 33–92 mg·g$^{-1}$), suggesting that the effect of these treatments on soil AN decreased over long-term manure application (Table 2). Moreover, the LT-$N_{min}$ rates differed between sites, fertilization treatments and time periods, suggesting that LT-$N_{min}$ was influenced by those diverse factors.

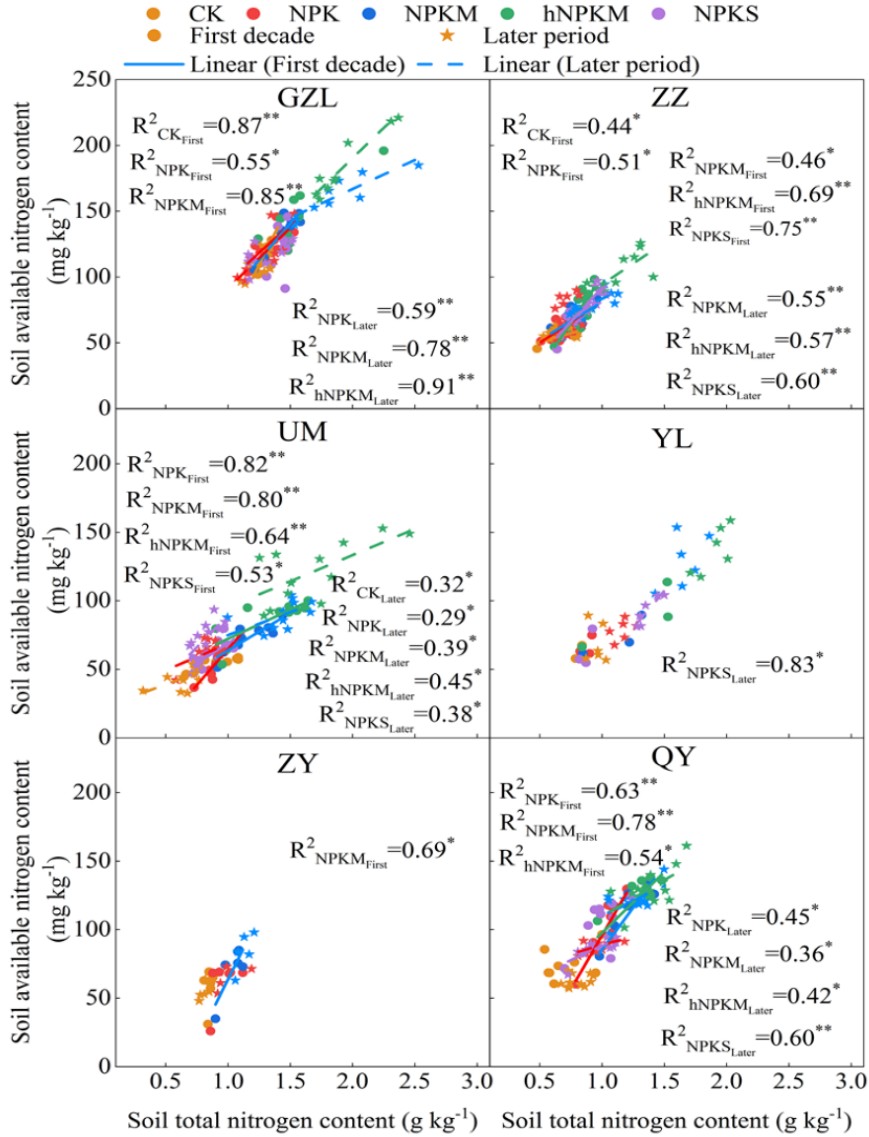

**Figure 3: The linear relationship between soil AN and TN content in the first decade and later period under various long-term treatments at the six study sites.** The slope of the linear relationship between AN and TN was used to calculate the annual increase in AN per 1 gram· TN ha$^{-1}$ increase annually. GZL, Gongzhuling; ZZ, Zhengzhou; UM, Urumqi; YL, Yangling; ZY, Zhangye; QY, Qiyang. **, significant correlation between the soil AN and TN at $P < 0.01$. CK, unfertilized control; NPK, inorganic N, phosphorus and potassium; NPKM, inorganic NPK plus manure; hNPKM, high application rate of NPKM; NPKS, inorganic NPK with stover return.



**Table 2. Soil LT-N$_{min}$ rates (mg·g$^{-1}$) in the first decade and later period under various long-term treatments at the six study sites.**

| Sites | Years | n | CK | NPK | NPKM | hNPKM | NPKS |
|-------|-------|---|-----|-----|------|-------|------|
| GZL | 1–10 | 9 | 190** | 86.9* | 110** | 72.6 | 65.1 |
| | 11–23 | 9 | 51.9 | 92.1** | 44.0** | 91.9** | 42.0 |
| ZZ | 1–10 | 10 | 26.6* | 62.9* | 58.2* | 142 ** | 105** |
| | 11–22 | 11 | 22.2 | 38.0 | 44.1** | 60.1** | 100** |
| UM | 1–10 | 10 | 14.8 | 105** | 54.4** | 42.4** | 58.7* |
| | 11–22 | 12 | 33.3* | 39.0* | 33.3* | 38.3* | 97.9* |
| YL | 1–10 | 3 | 132 | 85.1 | 49.6 | 50.1 | 212 |
| | 11–20 | 6 | -36.8 | 44.4 | 68.9 | 104 | 108* |
| ZY | 1–10 | 6 | 109 | 134 | 181 * | NA | NA |
| | 11–21 | 5 | 99.7 | 40.2 | 163 | NA | NA |
| QY | 1–10 | 9 | 22.2 | 157** | 122** | 88.7* | 75.2 |
| | 11–23 | 13 | 10.0 | 25.6* | 43.5* | 53.2* | 42.3** |

Notes: The LT-N$_{min}$ value was determined using a least squares linear regression: Y = a + bX, where Y is the soil AN content; X is the TN content; b is the slope (i.e., the change rate, reflects the increased AN every TN increased). GZL, Gongzhuling; ZZ, Zhengzhou; UM, Urumqi; YL, Yangling; ZY, Zhangye; QY: Qiyang. CK, unfertilized control; NPK, inorganic N, phosphorus and potassium; NPKM, inorganic NPK plus manure; hNPKM, high application rate of NPKM; NPKS, inorganic NPK with stover return. n, quantity of the available annual data during the long-term treatment period. NA, data not available because the treatments are not included. *, significant correlation at P < 0.05; **, significant correlation at P < 0.01.

In light of the above findings showing a gradual change in the dynamics of long-term N mineralization, we next sought to determine which factors drive the higher LT-N$_{min}$ in the two period of treatments. Variance portioning analysis (VPA) revealed that differences in soil properties, fertilizer regimes, and climatic conditions could explain the majority of variation (57%) between treatments in the first decade (Fig. 4a) and accounted for almost all of the variation (90%) in the later period (Fig. 4b). More specifically, soil properties had the highest influence, while fertilizer regime and climatic conditions contributed relatively less effect (*i.e.,* accounting for 36%, 6.6% and 19% of the total variation, respectively), which indicated that soil properties comprised the dominant determining factor in LT-N$_{min}$ dynamics. In the later period, the influence of soil properties on LT-N$_{min}$ increased to 45.1%, while that of climatic conditions and fertilizer treatment decreased to 8% and 1%, respectively. The interpretation rate for soil properties combined with fertilizer treatment was 24.8%, while soil properties and climatic conditions together accounted for 16.6% of the total variation in the later period. Collectively, these results suggested that the influence of climate on LT-N$_{min}$ was attenuated with increasing duration of fertilization, while fertilizer regime enhanced, indirect contribution on by altering soil properties.





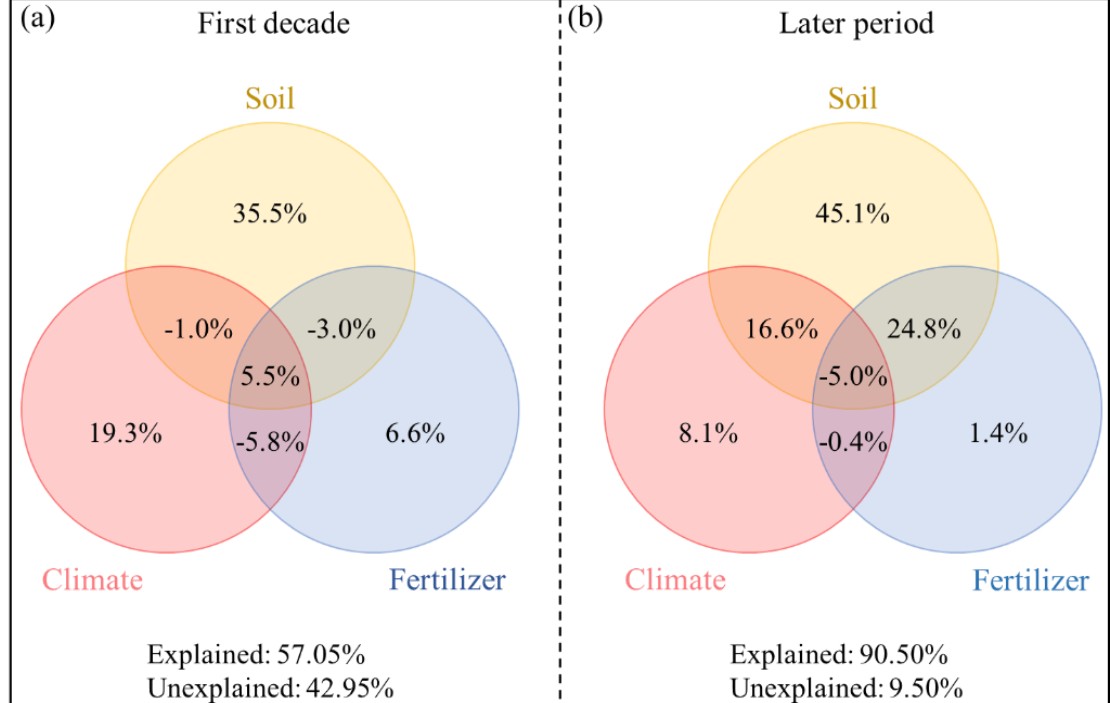

**Figure 4: Relative contributions of soil properties, climatic conditions and fertilization managements to LT-$N_{min}$ in the first decade (a) and later period (b).** Soil properties include pH, soil organic carbon (SOC), ratio of soil organic carbon and total N (C/N), total and available N, P and K nutrients contents. Fertilization managements include the unfertilized control (CK), inorganic N, phosphorus and potassium (NPK); inorganic NPK plus manure (NPKM), high application rate of NPKM (hNPKM), inorganic NPK with stover return (NPKS). Climatic conditions include mean annual temperature (MAT), mean annual precipitation (MAP), mean annual humidity and mean annual evaporation (AE). The total interpretation of soil LT-$N_{min}$ by were showed at bottom of the figures by soil properties, climatic conditions and fertilization managements.

### 3.4 Climate and long-term fertilization impact LT-$N_{min}$ via soil organic carbon, pH, nutrients and stoichiometry

We carried out structural equation modeling (SEM) to investigate the potentially contribution of soil properties, fertilizer treatments and climatic conditions to shaping LT-$N_{min}$ at these cereal production fields (Fig. 5a). In the first decade, the established SEM explained 32% of the total variation in LT-$N_{min}$ ($\chi2 = 5.565$; Fisher's C statistic $P = 0.591$, GFI = 0.955, IFI = 1.010, *RMSEA* = 0.000). In this model, soil nutrient content and stoichiometry both exhibited strong positive correlations with LT-$N_{min}$ (path coefficients = 0.69 and 0.90, respectively). In addition, mean annual temperature (MAT) and soil pH shared strong negative correlations with LT-$N_{min}$ (path coefficients = −0.86 and −0.70 respectively). Mean annual precipitation (MAP) indirectly affected LT-$N_{min}$ by impacting soil pH and nutrient stoichiometry, while fertilizer treatments shared an indirect positive correlation with LT-$N_{min}$ through its impacts on nutrient content, soil organic carbon (SOC) content, and nutrient stoichiometry. The high path coefficients for MAT and nutrient stoichiometry indicated that these factors provided the greatest direct influence on soil LT-$N_{min}$ in the first decade of treatment.

In the later period of fertilizer applications, the established SEM explained 34% of the total variation in LT-$N_{min}$ ($\chi2 = 4.508$; Fisher's C statistic $P = 0.875$, GFI = 0.958, IFI = 1.131, *RMSEA* = 0.000, Fig. 4b). In particular, SOC, nutrient content, and



MAT were negatively correlated with LT-$N_{min}$ (path coefficients = 0.69, 0.62, and 0.45, respectively). These high coefficient values indicated that SOC and nutrient content provided the strongest direct influence on soil LT-$N_{min}$ in the later period of the fertilizer applications. MAP was correlated with soil pH, but pH was not significantly correlated with LT-$N_{min}$. However, MAP indirectly affected LT-$N_{min}$ through impacts on SOC content. Similar with modeling results of data collected in the first decade,

fertilizer treatments showed no direct relationship with LT-$N_{min}$, but indirectly affected LT-$N_{min}$ by altering SOC and nutrient content. Overall, climatic effects (*i.e.*, MAP and MAT) led to decreases in LT-$N_{min}$ in the later period with fertilization. These cumulative data showed that the contributions of climatic conditions on LT-$N_{min}$ were alleviated over a decade by decreasing both direct influence and also its impact on soil LT-$N_{min}$ via pH.

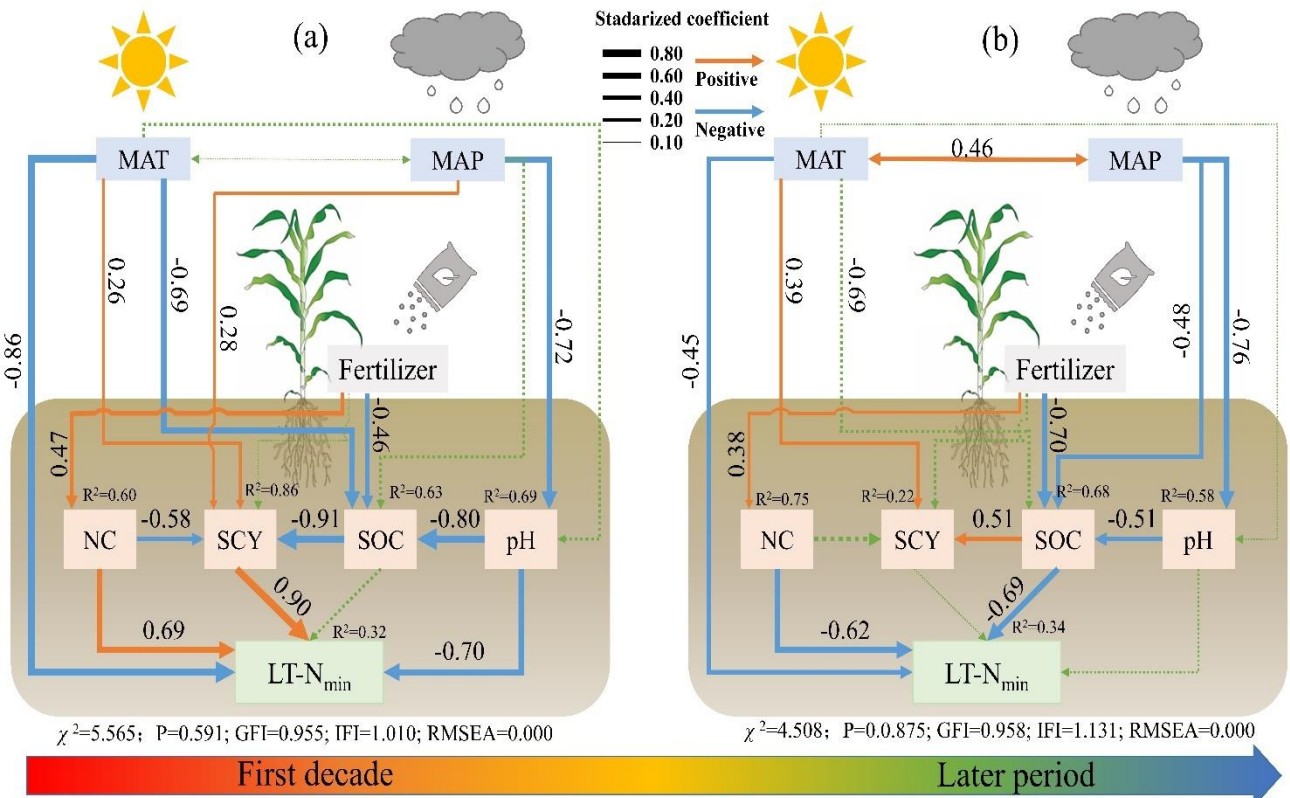

**Figure 5: Structural equation model (SEM) illustrating the direct and indirect effects of soil properties, fertilization managements and climatic conditions on soil LT-$N_{min}$: at the first decade (a) and the later period (b) of fertilizer applications.** Single-headed arrows indicate the hypothesized direction of causation and two-headed arrows indicate related correlations. Continuous and dashed arrows represent the significant and non-significant relationships, respectively. Orange and blue arrows indicate positive and negative relationships, respectively. The numbers adjacent to the arrows are the standardized path coefficients and the width of the arrow is in proportion to the

degree of standardized path coefficients. $R^2$ values indicate the proportion of variance explained by each variable. The nutrient content (NC), stoichiometry (SCY) and fertilizer represent the first component from PCA conducted for soil total and available nutrient of N, P and K; C/N, C/P, N/P; and the treatments of CK, NPK, NPKM, hNPKM and NPKS, respectively. The chi-square ($\chi^2$), nonsignificant probability (P > 0.05), high goodness-of fit index (GFI > 0.90), high incremental fit index (IFI > 0.90), and low root-mean-square errors of approximation (RMSEA < 0.05) listed below the SEMs indicate our data matches the hypothetical models. MAT, mean annual temperature; MAP, mean

annual precipitation.





## 4    Discussion

We found that manure application enhanced both TN and AN accumulation in soil (Table 1). The effects of manure on improving soil TN retention have been confirmed in previous studies (Li and Shi, 2007; Gao et al., 2005). However, the role of manure in N availability was less clear or AN data were often inconsistent due to its high variability under different environmental conditions. To our knowledge, this study represented the first estimation of soil long-term N mineralization (LT-$N_{min}$), and showed that enhanced N availability was consistently coupled with increased TN, 7–8% of which could be transformed into available N (Fig. 2). It should be noted that LT-$N_{min}$ values at all sites were higher in the first decade than those in the later period in plots treated with manure (Fig. 3 and Table 2). This finding indicated that soil organic N tends to be mineralized in the early stages of manure fertilization, and thus the mineralization rate declined over prolonged manure application. One explanation for this difference in LT-$N_{min}$ between decades was that manure application significantly increased soil C and N contents in the first decade, but slowed down in later stages (Gong et al., 2011; Spargo et al., 2011). Other studies have shown that soil carbon adsorption reaches saturation with increasing experimental time due to the decreased soil-specific surface area and adsorption sites by manure application (Klaus and Georg, 2000). This decrease in adsorption could result in limited microbial growth, which may have ultimately alleviated N mineralization in the later period (Mohanty et al., 2013).

Another explanation for the observed decrease in LT-$N_{min}$ was that repeated fertilizer application led to high $NH_4^+$ content, which was widely known to repress the activity of enzymes responsible for breaking down organic polymers (Padhan et al., 2020), and thereby decreasing LT-$N_{min}$ in the later period. Furthermore, repetitive manure application could potentially provide direct substrate to stimulate autotrophic nitrification, consequently enriching the size and activity of ammonia-oxidizing bacterial and archaeal population in these soils; the enhanced autotrophic nitrification may have led to higher N loss (Marchant et al., 2016). These results thus provided a framework for fertilizer management to maximize N use efficiency and N retention in agricultural soils.

Variance portioning analysis showed that soil properties were the primary contributing factor to LT-$N_{min}$ rates (Fig. 4). Among the relevant soil parameters, pH, SOC, nutrient stoichiometry, and nutrient content influenced directly or indirectly LT-$N_{min}$ in both decades (Fig. 5). However, the modes by which these soil properties contributed to soil LT-$N_{min}$ differed between the first decade and later period. Soil pH was widely considered a dominant selection factor for soil microbes, as well as for metabolic and enzymatic activities in soil, which was directly correlated with LT-$N_{min}$ in the first decade but indirectly affected LT-$N_{min}$ via SOC in the later period (Fig. 5). Other studies have proposed that a narrow pH range in later stages of treatment indirectly affect soil microbial community composition (DeForest et al., 2012). Nutrient content was positively correlated with LT-$N_{min}$ in the first decade, but negatively correlated in the later period, suggesting that the mechanisms mediating its effects on LT-$N_{min}$ changed in the later stage. In low concentrations, soil nutrients supported microbial growth and promoted LT-$N_{min}$. However, high nutrient accumulation promoted soil N immobilization, which resulted in decreasing N mineralization (Choi et al., 2004), and provided an alternative source of N which consequently lowered crop requirements for mineralization of soil N (Meyer et al., 2018). Therefore, it was important to clarify the direct and indirect influences of various soil parameters on LT-



N_min to better understand the process of soil N transformation and regulation of available N for crops.

Our models showed that temperature and precipitation also provided a stronger contribution to LT-N$_{min}$ in the first decade (19%) than in the later period (8%) (Fig. 4). This finding suggested that climatic impacts on soil LT-N$_{min}$ could be attenuated by long-term fertilization practices. This possibility was supported by our results showing that MAT directly affected LT-N$_{min}$ with a higher standardized path coefficient (0.86) in the first decade than in the later period (0.45) (Fig. 5). Notably, annual temperatures were higher in the later period than in the first decade at all sites except GZL, increasing by 0.7°C on average

across the six sites (Table S4). There are three primary modes through which climate can affect LT-N$_{min}$. First, temperature and moisture significantly affected the soil microbial biomass and activity. In a previous study, Liu and colleagues (2016) calculated Q10 values to assess the temperature sensitivity of soil N mineralization and found that higher Q10 was associated with increased soil mineralization rates. Second, precipitation significantly and positively influenced N deposition on a global scale (Serna-Chavez et al. 2013), and detection of microbial carbon suggested that N deposition could decrease soil microbial

biomass production by 15% (Treseder, 2010) which in turn inhibiting LT-N$_{min}$. Third, climatic conditions affected LT-N$_{min}$ by altering soil properties, especially pH and SOC in our study.

In both periods with fertilization, annual precipitation was correlated with pH, while pH was negatively correlated with LT-N$_{min}$ in the first decade but had no relationship with LT-N$_{min}$ in the later period (Fig. 5). Soil pH has been described as the second most important factor affecting N$_{min}$ on a global scale (Li et al., 2020), although the relationship between soil pH and

N mineralization was inconsistent among published studies. Increasing soil pH from 3.8 to 6.8 was shown to influence both soil gross N mineralization and immobilization rates, resulting in declining trend in N$_{min}$ rates (Cheng et al., 2013). By contrast, Fu et al. (1987) reported that N$_{min}$ increased with the increase of soil pH due to the promotion of substrate availability by high pH conditions. Kemmitt et al. (2006) found that both soil N mineralization and nitrification rates were positively correlated with pH, since soil acidity limited soil microbial activity. The contrasting effects of pH on N$_{min}$ may be related to differences

in the effects of soil pH on gross N mineralization and gross N immobilization, because microbial populations were responsible for both the release and immobilization of soil nutrients. It is worth noting that variation in pH was higher in first decade than in later period for most treatments in this study due to pH had a negative relation with MAP (Fig. 5 and Table S5), which could also contribute to the variability of climatic influence on LT-N$_{min}$ under long-term soil management regimes.

Temperature and precipitation were both independently correlated with SOC content, which was correlated with LT-N$_{min}$ in

both decades, suggesting that the SOC response to climatic conditions was a determining factor in LT-N$_{min}$. The increase in annual temperature recorded in the later period could potentially drive the transfer of labile C into soil by increasing the decomposition rate of SOC, which subsequently enhanced microbial N demand and decreases LT-N$_{min}$ (Rustad et al., 2001). In our study, trends in SOC content differed between the first decade and later period potentially due to different treatments (Table S6). For example, SOC significantly increased under manure application at the YL and QY sites during the first decade,

but did not significantly change during the later period, which could lead to variability in LT-N$_{min}$. Furthermore, SOC content indirectly affected LT-N$_{min}$ by shaping soil nutrient stoichiometry in the first decade, but directly impacted LT-N$_{min}$ in the later period (Fig. 5), suggesting that the mechanisms by which SOC affected LT-N$_{min}$ changed over time. These results thus



illustrated how climate directly modulate soil LT-$N_{min}$ and shape its response to various soil conditions. Li et al. (2019) reported that 30% of the unexplained variations in $N_{min}$ may be attributable to shifts in microbial community structure. However, it
remained unclear how microbial communities impact $N_{min}$ due to the complexity and diversity of metabolic processes and environmental conditions at play in soils. A comprehensive understanding of the regulatory mechanisms by which climatic conditions and soil properties modulate LT-$N_{min}$ at different times could quantify the long-term residuary effects and mechanism of fertilization on soil N availability, which would facilitate the establishment of management strategies appropriate for specific soils and growing conditions to maximize crop N utilization.

**5      Conclusions**

Evaluation and SEM modeling of soil N under conventional fertilizer or conventional fertilizer plus manure treatments in 6 soils > 20-year field research sites showed that both total soil N and available N contents were significantly increased by application of manure except in one acidic soil. Total soil N content shared a positive linear correlation available N content, and the slope of the linear regression (*i.e.,* LT-$N_{min}$, the long-term N mineralization rate) indicated that available N increased
by 70–81 mg per gram total N increased in soil. The LT-$N_{min}$ rate was higher in the first decade of fertilizer treatments than in the later period. Mean annual temperature, soil pH, nutrient contents, and stoichiometry were shown to be determining factors that affect LT-$N_{min}$ rate, and may drive major differences over long time periods. Climatic conditions also strongly contributed to LT-$N_{min}$ rates in the first decade, but decreased the effect in the later period. However, the effects of climate interactions with soil properties on LT-$N_{min}$ rates were enhanced in the later period. Collectively, these results suggested that manure
incorporation with conventional N fertilizers could be a highly effective strategy to improve soil fertility/productivity for high crop yield while also alleviating the effects of climate variation on soil N availability.

**Author contributions**

Y.H.D, and M.G.X designed the study, Y.H.D, M.G.X, and H.Q.Z developed the overall research idea and wrote the first draft, Y.H.D, M.G.X, W.J.Z, and C.A.L revised, discussed, and finalized the manuscript. H.Q.Z, K.Y.R, and D.J.L helped in data
analysis. All authors contributed to the manuscript writing, discussion, and revision.

**Funding**

This work was supported by the National Key Research and Development Program of China [No. 2021YFD1900300]; National Natural Science Foundation of China [No. 42077098]; and Fundamental Research Funds for Central Non-profit Scientific Institution [No. Y2022XK26].



**Data availability**

Data generated in this study can be made available upon request from the corresponding author.

**Declaration of Competing Interest**

The authors declare that they have no known competing financial interests or personal relationships that could have appeared to influence the work reported in this paper.

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
