# Peer review of "Discrepant long-term nitrogen mineralization in soil at early and later period after fertilization"

_EGUsphere, 2024_

## Author Comment (AC1)

Dear Editor and Reviewers,

Thank you for the careful review and insightful feedback on our manuscript "Discrepant long-term nitrogen mineralization in soil at early and later period after fertilization" (egusphere-2024-330). We are grateful for the two anonymous reviewers for the many constructive and valuable comments for us to improve the quality and clarity of our manuscript. These reviews made us realize that we missed some key information or not justify many points well that have caused many concerns or confusions. We believe we can significantly improve the paper and can address each of the comments in a revised version of the manuscript. One of the main reasons is that the paper reports findings from long-term field experiment up to ~ 30 years and from different geographical locations and no such studies existed in our knowledge; thus, it will fill many knowledge gaps regarding what effects of long-term organic (manure in this case) amendments as a nutrient source have on soil carbon, nitrogen and other nutrients and implications on soil ecosystem functions, nutrient management, and environmental aspects. We realize that we can achieve these by making our best efforts, which was not done well previously. Thus, we would like to request sometime to address the reviewers' comments and be able to achieve much more with the data. In the revision, we will ensure to provide a clear rationale to elucidate the significance of long-term N mineralization study, an abstract that is precise and substantive, a detailed exposition of the experimental design and its significance, examine the data, and synthesize all information for reporting results clearly, making meaningful discussions, and drawing concrete conclusions. We will address the English writing and grammatical issues. Our response or intent to revise the paper are listed below in dark-blue text following each comment:

Dear Editor,

I understand this manuscript proposes how the calculation of "long-term" N mineralization rates is important and that most studies have focused on "short-term" rates. While this may be the case, the manuscript does not develop a clear rationale as to why these long-term rates are needed and how they will help fill a critical knowledge gap in our understanding of N cycling. Thus, I am unable to recommend publication of

this manuscript in its current form as it requires revisions that go beyond a major revision. Systematic unclarity in the writing and grammatical issues makes the quality of the science difficult to assess.

Thanks for this critical comment for the paper. To be brief here, the rationale as to why these long-term N mineralization rates are needed is mainly because we have been promoting nutrient cycling such as using manure in crop production, although most conventional farmers are still using mainly synthetic fertilizer for different reasons. Nitrogen dynamics is an important feature that affect soil nutrient statues and management decision making. Many studies on N mineralization rates are conducted in the lab or short-term. We realized that our long-term experiments in multiple geographical locations can bring some new (as no one has done this before) if we can examine them well. Although we do realize that large field variability and multiple environmental/geographical variable may not make it easy as some of the comments have been made on this, we believed it is a worthwhile attempt as this new knowledge will benefit us all for better understanding of N cycling, pros and cons of the different fertilization regimes, and further guide our decision-making. We will add this information and organize better in the introduction to justify the importance why we conducted this research. The new knowledge can guide further nutrient management practices, especially in using both organic and mineral fertilizers for the intensification of cropping systems under long time scales.

We also acknowledge the need for improvement in the clarity of our writing and grammatical issues. We will seek for professional editing help and/or introduce some experts in the research field and also English-speaking professional to help revise the manuscript so that we will ensure that the scientific content will be presented in a clear and coherent manner, facilitating a more accurate assessment of the quality of the research.

Because the manuscript does not contain continuous line numbers, it is difficult to provide a detailed review. Below I highlight the range in which specific issues require attention

We will make sure to use line numbers for each line in the revised manuscript.

Specific comments (not an exhaustive list):

1. Title, I suggest editing the title because "early" and "later" period are not concrete and it is unclear what timeframe they represent

We will replace the "the first decade" and "the later period" with more common used and concrete terms such as "the first ten years" and "after 10 years" in the full text. And we will revise the title "Discrepant long-term nitrogen mineralization in soil at early and later period after fertilization" to be more straightforward such as "Changes in soil nitrogen mineralization rate with time from long-term fertilization".

2. Abstract, Not accurate. First, mineralization covers many elements, not just N. Second, N mineralization calculations also include the conversion of $NH_4$ to $NO_3$. Third N mineralization does not always balance crop N uptake and N loss

We will revise the sentence to "Soil nitrogen (N) mineralization is a process in the N cycle, which converts organic N into more plant available inorganic form ($NH_4^+$ and $NO_3^-$), improving N availability for crop growth and also causing potential loss to the environment."

3. Unclear what is a "residuary" effect. Readers are assumed to know this term and I recommend not assuming this.

We will replace the "residuary" effect to "residual" effect and include an explanation of the term "residual effect". The residual effect refers to the circumstance wherein immobilized nitrogen and $NH_4^+$–N fixed to mineral and organic components may not be readily available to the crop immediately after the application of fertilizers (Sørensen and Jensen, 1995; Sørensen and Amato, 2002) while can be utilized by subsequent crops, a phenomenon known as the residual effect.

4. A case has not been built to support the idea that studying manure application is important.

We will address this concern by providing the rationales as indicated in the response to

the general concern. From quick thoughts, we will include the following: 1) The importance of understanding of N dynamics from mineralization in sustainable crop production by enhancing nutrient recycling, 2) the necessity for meeting crop nutrient requirements, 3) the need for developing best management practices etc. All these are associated with mineralization of manure. Further, field especially long-term experimental data have been rarely studied in this regard although they can provide the most needed net outcome information, which is the strength of this research.

5. Unclear why this particular experimental design is required to answer the question this manuscript is set to answer, which is also difficult to understand. We are told that studying long-term N mineralization is important, but we have not been told why.

The responses to the general comments and that to #4 have addressed most of this concern. From long-term fertilization experiments, we found both total N and available N continuously increased, understanding their relationship and proportional changes directly links to soil property and nutrient availability. We know that immobilized nitrogen and $NH_4^+$–N fixed to mineral and organic components may not be readily available to the crop immediately but overtime they can be utilized by subsequent crops. But how much and how fast are what we try to understand, using the calculated annual N mineralization rate by the annual changes of available N to that of total N from the long-term field experiment, that we defined as the long-term N-min rate. Further, these changes over time with varying fertilizers across diverse climate zones and soil types remains unclear. Gaining insight into this phenomenon is beneficial for guiding nutrient management practices to enhance soil fertility, especially when both organic and mineral fertilizers are increasingly practiced for the intensification of cropping systems.

6. L30, I recommend rewriting and updating references as it is now well known that many plants can take up a significant amount of organic N. Also what is "enhanced soil N mineralization" and how does it differ from just N mineralization? I recommend defining N fertilizer use efficiency. Because everything leading to the end of the sentence is unclear, it is also unclear why this enabled sufficient crop yields with reduced risk in N loss.

We totally agree with the comments and will update the references and improve writing accuracy and clarity. We will definitely revise "enhance soil N mineralization" and also the whole sentence to make it clear to understand logically.

7. The use of "therefore" is confusing because the information in the prior sentence does not lead into this section after "therefore."

We will replace the "therefore" with something more appropriate such as "underscoring": "Consequently, plants still rely on the mineralization of organic to inorganic N to meet their nitrogen requirements for growth and maximize crop biomass production (Giacomo et al., 2012), underscoring the significant impact of soil N mineralization on crop production."

8. L40, this is a critical part of the manuscript where the authors must make it very clear why this research is important and why determining long-term N mineralization is critical. It remains unclear why this is important and the introduction needs to build a strong case for this.

We totally agree with the reviewers and truly appreciate that the reviewer mentioned several times about missing such critical information in the previous version of the manuscript. As indicated in the response to the general comments and that to #4, we will be able to add the rationales for this research. We will revise the whole introduction section to address these points and build a strong case for the importance of this research.

9. What are the long-term fertilization effects that occur under field conditions?

We will revise this term or specify it in the revised manuscript, such as "Estimating site-specific N mineralization rate from the long-term field experiment will capture the differences in changes (increase or decrease) in supply of available nutrients levels that consequently impact decision making on fertilization needs.".

10. L45, the rationale included in these lines is difficult to follow. In particular, it must be clarified what does it mean that the inorganic mineralization substrates are

included in the total N pool and why does this mean that total N would be a good predictor. Then after this sentence we are told often times these are not good predictors. This is confusing.

These were not written clearly apparently. We will revise the sentences to improve clarity.

11. L65, the background in the introduction that would be necessary to understand this objective is not clear, and it is not clear why this information is critical and why the experimental design used is the right approach to answer the question this study is set to answer and how it will advance the field of N biogeochemical cycling.

We again truly appreciate the reviewer's review and acknowledge the need to provide clear rationales in the introduction to help understand why we carried out this study, the objectives, methods used, and implications of the findings. See response to the general comments and that to #4.

---

## Author Comment (AC2)

Dear Editor and Reviewers,

Thank you for the careful review and insightful feedback on our manuscript "Discrepant long-term nitrogen mineralization in soil at early and later period after fertilization" (egusphere-2024-330). We are grateful for the two anonymous reviewers for the many constructive and valuable comments for us to improve the quality and clarity of our manuscript. These reviews made us realize that we missed some key information or not justify many points well that have caused many concerns or confusions. We believe we can significantly improve the paper and can address each of the comments in a revised version of the manuscript. One of the main reasons is that the paper reports findings from long-term field experiment up to ~ 30 years and from different geographical locations and no such studies existed in our knowledge; thus, it will fill many knowledge gaps regarding what effects of long-term organic (manure in this case) amendments as a nutrient source have on soil carbon, nitrogen and other nutrients and implications on soil ecosystem functions, nutrient management, and environmental aspects. We realize that we can achieve these by making our best efforts, which was not done well previously. Thus, we would like to request sometime to address the reviewers' comments and be able to achieve much more with the data. In the revision, we will ensure to provide a clear rationale to elucidate the significance of long-term N mineralization study, an abstract that is precise and substantive, a detailed exposition of the experimental design and its significance, examine the data, and synthesize all information for reporting results clearly, making meaningful discussions, and drawing concrete conclusions. We will address the English writing and grammatical issues. Our response or intent to revise the paper are listed below in dark-blue text following each comment:

Dear Editor,

It is evident that understanding about soil N turnover must be improved, especially for agricultural systems to drive it towards sustainability path. Here authors have shown the long term effect of fertilization especially, organic fertilization increases nitrogen availability and a complex interaction of climate and soil variables affect LT-$N_{min}$,

different over different period of time. Based on following reasons I suggest to reject the paper in its current format.

1. All sites are in different location and posses different fertilization regimes. The complexity of it might have considerable effect on TN and AN and eventually, LT-$N_{min}$. For e.g, basal fertilization seems not done in all sites and it can highly impact on AN especially because AN is highly dynamic and in the absence of plant there are high possibility of it leaching underground and or to surface water mainly due to precipitation event. Additionally, the exact soil timing along with field conditions has not been reported which make difficult to asses the difference between sites. A sign of unclear method presentation.

We agree that the different study sites with many varying factors (locations, climatic conditions, and fertilization practices, etc.) add difficulties and challenges for exact same treatments being applied, type of crops, processes that impact the fate of N etc. However, the differences in the study sites also bring us the advantages to reveal similarities and differences in the outcome of the same treatment or approach in replacing mineral N fertilizer with manure. We should have made it clear that the same treatment at the different sites were not exactly the same in nutrient application amount and time, but fertilization rates were based on the common practice locally used rates for high yield, which were from years of experience and served as the base of application rate. Then the same treatments at the different sites referring to the same proportion of organic (manure) N source. In this way, it will make the specific numbers and when the fertilizers used such as base or no base less relevant.

Further all soil samples were collected immediately after harvest in the autumn, around end of September to early October, depend on the site. We apologize for not making this clear in the previous version of manuscript, hope the reviewer agree with our reasoning, and will make sure to significantly improve the methodology for clarity. To summarize, the soil AN and TN content used in the estimation of LT-$N_{min}$ were obtained after harvest in the fall, which provide the N status at the end of the plants growing cycle, enabling an evaluation of changes (accumulation or depletion) affected by fertilization regimes that allows us to examine long-term effects on soil N availability. We will do our best to revise the method presentation to ensure clarity and coherence.

2. Agreeing with the previous reviewer, the use of term "residual N" is difficult to grasp. An assumption has been made and reported in the abstract but introduced in method.

We will revise the text to include an explanation of the term "residual N" such as: we will replace the "residuary" effect to "residual" effect, and include an explanation of the term "residual effect". The residual effect refers to the circumstance wherein immobilized nitrogen and $NH_4^+$–N fixed to mineral and organic components may not be readily available to the crop immediately after the application of fertilizers (Sørensen and Jensen, 1995; Sørensen and Amato, 2002) while can be utilized by subsequent crops, a phenomenon known as the residual effect.

3. Can really a slope between TN and AN represent the true LT-$N_{min}$ and with two data points over a one year period? especially when we know AN is highly dynamic and it is even more difficult to judge if background information (sampling timing, field conditions) about AN assessment is lacking.

We agree that long-term nitrogen mineralization may not be exactly represented by the slope between TN and AN. This research is based on our observation from the long-term fertilization experiments where we observed an increase in both soil total nitrogen and available nitrogen across multiple years. We reasoned that AN changes are from mineralization of organic matter which can be presented by the increased TN from manure application. Consequently, we leveraged multi-year soil AN and TN data to establish their linear regression relationship, and defined the slope as LT-$N_{min}$, which could reflect how much TN was converted into AN during the accumulation process of soil nitrogen.
The sample time and reasons why were explained earlier. Basically, assuming all other processes are resulted in similar fate of pathway, the AN determined represented the net change in soil AN. The assumptions and explanations will be added to methodology.

Other aspects that i found critical are listed below:
1. "Nutrients (N, P, and K) in the stover, however, were not counted towards N applications because changes in their availability with time were not recorded." While is is understandable about the difficulty to estimate the NPK in stover over a long period of time because or resources but it is scientifically correct to accept that

they could have contributed substantially to soil TN and AN and not accounting them could lead to incorrect estimation of TN and AN.

We agree that stover return does bring a lot of nutrients (N, P, and K) into soil over a long period of time. In this study, we just didn't count them in the nutrient input in fertilization amount calculations. The measured total and available nitrogen in the soil after crop harvest does include their effects. We will clarify this and indicate possible additional amount of nutrients were introduced from the stover return treatments and incorporate the findings in our discussion.

2. The method section could have been improved a lot, especially regarding soil sampling, fertilization regimes and sample handling. All the sites at different location has different management practices and a clear and easy to follow text would have been very much helpful. Also, author could have considered presenting method that are basic in the first place, for e.g., plot design, size, and number and then with more detail about experiment.

Yes, we have realized the need for significant improvement in the method section as the reviewers indicated. We will provide detailed description of soil sampling, fertilization regimes, and sample handling procedures, plus reasoning and possible effects on results etc. We will reorganize the study method by initially outlining fundamental aspects such as plot design, size, and number, followed by a more comprehensive description of the experiment.

3. Differences in soil LT-$N_{min}$ between the first decade and later period of fertilizer application and the factors influencing these changes. Here is the first time that i am seeing the use of term "first decade" and "later period". I think a suffcient information about what they are and which period of time they refer to should be in method section.

The experiments were conducted at different location and also during different time periods, some started in 1982 and others in 1990. We segmented the experimental data from 21 to 29 years into two phases: the first 10 years and after 10 years, which are the common terms most would accept we think. The reason is that we observed significant shift in the LT-$N_{min}$ during those two different time frames. We will revise it and make

the explanation about it in the revised manuscript.

4. the enhanced autotrophic nitrification may have led to higher N loss, is a N loss can be used as framework for fertilizer management to maximize N use efficiency and N retention in agricultural soils. i am not sure about it.

These were not written clearly apparently, and we will revise it.

Therefore, my suggestion is to reject the paper in its current form mainly because of the complexity with fertilization regime which are very different in all sites and soil sampling timing which is not provided in a clear manner and I believe they both can have considerable impact on TN and AN leading to uncertain estimation of LT-N$_{min}$, and thus to incorrect conclusions.

We truly appreciate the in-depth review by the reviewers that have made us realize several key problems with previous versions of the manuscript. Our apology for the frustration that reviewers might have experienced while reading our paper. We are confident, however, about addressing all the issues raised by the reviewers. We will carefully address each of the comments in our revision. We will reexamine the paper holistically to improve its overall quality, present the scientific information clearly and accurately, and hopefully deliver some scientific findings that help advance understanding of the fate of N under different fertilization regimes that can be used in management.